# Bioinformatics in Plant Breeding and Research on Disease Resistance

**DOI:** 10.3390/plants11223118

**Published:** 2022-11-15

**Authors:** Huiying Mu, Baoshan Wang, Fang Yuan

**Affiliations:** Shandong Provincial Key Laboratory of Plant Stress, College of Life Sciences, Shandong Normal University, Jinan 250014, China

**Keywords:** plant resistance, plant breeding, bioinformatics, automated robots, precision fertilization

## Abstract

In the context of plant breeding, bioinformatics can empower genetic and genomic selection to determine the optimal combination of genotypes that will produce a desired phenotype and help expedite the isolation of these new varieties. Bioinformatics is also instrumental in collecting and processing plant phenotypes, which facilitates plant breeding. Robots that use automated and digital technologies to collect and analyze different types of information to monitor the environment in which plants grow, analyze the environmental stresses they face, and promptly optimize suboptimal and adverse growth conditions accordingly, have helped plant research and saved human resources. In this paper, we describe the use of various bioinformatics databases and algorithms and explore their potential applications in plant breeding and for research on plant disease resistance.

## 1. Introduction

Broadly speaking, bioinformatics covers the interdisciplinary studies of biological objects (including genes, proteins, and physiological indices) using informatics methods, such as various algorithms and statistical methods. Specifically, complex biological data can be processed using computer tools, which is common practice in dedicated databases, such as in nucleic acid databases, protein databases, and custom functional databases [1]. The implementation of bioinformatics tools reduces the cost of complex analyses, thus enhancing research into topics such as sustainable agriculture [2]. Understanding how bioinformatics can be applied to plant biology research is therefore important for researchers in the life sciences, and here, we have provided a description of these tools and their applications, focusing on plant breeding and research on disease resistance. For example, based on the VPg gene sequence of a PVY (Y virus) isolated from potato, combined with all published sequences in GenBank, two things can be inferred: the rate of evolution of PVY and the time to reach the most recent common ancestor using a Bayesian system dynamics framework to advance disease resistance studies in potatoes [3,4,5]. Given that multifactorial traits involved in resistance and quality are extremely difficult to improve, especially in combinations, and some of the genomes of major forage crops, such as maize, rice, wheat, sorghum and barley, and the forage legumes soybean and alfalfa, are too large to be analyzed using whole-genome sequencing, attention has been focused on comparative genomic approaches in order to produce seeds with desirable shapes [6,7,8].

The typical datasets generated by plant researchers contain morphological, physiological, molecular, and genetic information that describes the entire plant life cycle. Bioinformatics process the collected data and extract key indices and trends to quickly and accurately generate hypotheses and then offer solutions. For example, phenotypes and genotypes can be combined to reveal the underlying mechanism, such as the study of plant rejuvenation [9], and the future growth pattern of plants can be predicted according to the growth trend of plants in the past, such as the plant growth pattern prediction system, developed by deep learning [10], and the comparison of multiple genomes can be used for the prediction of evolutionary relationships, such as in the study of *Amphicarpaea edgeworthii* [11].

In agricultural applications, the wide utilization of bioinformatics can assist with efficient crop breeding and the improvement of plant resistance against pathogens [12]. In particular, scientists are committed to breeding and modifying crop species to improve the yield and quality, as well as creating new varieties with qualities that benefit human nutrition and health. Bioinformatics accelerates the generation and deployment of these new varieties. Indeed, genes associated with specific traits can be analyzed on a computer before being introduced into a plant, and the results can be used to determine what to introduce further into the plant for a precise phenotypic analysis. Maize (*Zea mays* L.) kernels, rich in lysine [13]; lettuce (*Lactuca sativa*), high in vitamin C [14]; and the recently developed vitamin D-rich tomato [15] are examples of the implementation of such pipelines.

Bioinformatics plays a critical role in data integration, analysis, and model prediction, as well as in managing the massive amounts of data resulting from new, high-throughput approaches [16]. Classical biological experiments, such as the visualization of mitosis and meiosis and pollen tube growth, are undergoing deeper, higher throughput exploration thanks to bioinformatics and time-lapse microscopy [17,18]. Plant growth can be predicted based on the available wealth of physiological and phenotypic data, enabling the generation of a virtual plant that can accurately predict growth patterns and the consequences of interactions with diseases or pests [19]. Bioinformatics has also wide applications in the analysis of plant resistance to various stresses [20]. The molecular mechanisms underlying plant responses to abiotic stress have been studied in depth, and they can open new avenues in agriculture when combined with the predictive power of bioinformatics [21]. In addition, bioinformatics has been applied in plant pathology, such as identifying and predicting the “effector” proteins produced by plant pathogens in order to manipulate their host plants. The functional annotation of this pathogen’s ability to predict virulence is a critical step in translating the sequence data into potential applications in plant pathology [22]. A bioinformatics framework has been proposed to enable stakeholders to make more informed decisions. In this way, a shared biosecurity infrastructure can be established to cater for sustainable global food and fiber production in the context of global climate change and the increased chances of accidental disease invasions in the global plant trade [23]. To develop new strategies for plant disease control, researchers must elucidate the complex molecular mechanisms underlying pathogen infection. Whole genome sequencing technology has enabled the sequencing of an increasing number of pathogens and the accumulation of large amounts of genetic data. Therefore, bioinformatics tools for analyzing pathogen genomes, effectors, and interspecific interactions have been developed to understand disease infection mechanisms and pathogenic targets, which all contribute to plant pathology [24].

In this review, we focus on the applications of bioinformatics in crop breeding and the study of resistance to various stress factors: to (1) list the applications of bioinformatics in plant research, (2) to clarify the application of bioinformatics in plant breeding, (3) to emphasize the advances made by bioinformatics in the study of plant tolerance and disease resistance, (4) to predict the plant growth by bioinformatics, and (5) to call for a greater use of bioinformatic methods in plant research. 

## 2. Databases Provide Abundant Gene and Pathway Information to Study Plant Biology

Thanks to large-scale sequencing technologies, vast amounts of data are released continuously and are often uploaded to a specific database. Depending on the species they represent, databases can be formally classified as general or species-specific databases.

General databases include those that integrate information about genomes, proteins, and metabolic pathways (Table 1). Genome databases represent a centralized and public collection of all published data, so researchers can easily obtain information concerning their gene or protein of interest. For example, UniProt offers a comprehensive resource for protein sequences and functional annotation. The database can be queried with a specific gene/protein name or with keywords of interest to sort through the catalogued data, but it is also possible to perform a protein BLAST (basic local alignment search tool) and download the sequence of the new protein of interest [25]. In addition, general databases compile various biological pathways, such as those represented in Gene Ontology (GO), Kyoto Encyclopedia of Genes and Genomes (KEGG), EuKaryotic Orthologous Groups (KOG), and metabolic pathways, which can be used to determine if a candidate protein belongs to one of many known pathways.

As one example, a bioinformatic analysis of ribulose-1,5-bisphosphate carboxylase/oxygenase (RuBisCO) from multiple C3 plant species included an in silico characterization of RuBisCO and its interacting proteins, whose structures and functions were predicted with the ProtParam, SOPMA, Predotar 1.03, SignalP 4.1, TargetP 1.1, and TMHMM 2.0 tools, which are all accessible from the ExPASy database. A MEME and MAST analysis of RuBisCO from all C3 plants, combined with a phylogenetic tree constructed with MEGA 6.06 software based on a sequence alignment obtained with the ClustalW algorithm, illustrated the high-sequence identity shared by RuBisCO from different C3 plant species, supporting the notion that they originated from a common ancestor [26]. A list of these databases and how they are used is provided in Table 1.

The model plant *Arabidopsis thaliana* is one of a few plant species with its own databases due to its widespread use in plant research (Table 2). These databases are rich in resources and can help researchers quickly obtain the latest Arabidopsis genome information. For example, The Arabidopsis Information Resource (TAIR) database allows users to download gene sequences in bulk, while the SeqViewer in TAIR also provides a simple tool to visualize the genes. In addition, TAIR has a powerful function for displaying various expression maps, each representing expression data during Arabidopsis development or under different growth conditions [27].

Most major crops have dedicated databases, including rice (*Oryza sativa*), wheat (*Triticum aestivum*), barley (*Hordeum vulgare*), maize, soybean (*Glycine max*), cotton (*Gossypium hirsutum*), and sorghum (*Sorghum bicolor*) (Table 3). For example, the Rice Mutant Database (RMD) includes mutants for the identification of new genes and regulatory elements and includes a list of lines for the ectopic expression of target genes in specific tissues or at specific growth stages, providing rich data resources for the study of different rice mutants [28]. The Wheat Genomic Variation Database (WGVD) compiles all published single nucleotide polymorphisms (SNPs), insertion/deletion polymorphisms (InDels), and selection sweeps, together with a BLAST search tool for wheat [29]. Researchers at Huazhong Agricultural University have developed the ZEAMAP database for corn, which includes multiple omics data resources, such as genomes, transcriptomes and genetic variation, phenotypic data, metabolome studies, and genetic maps. The database also provides access to a variety of data on complex traits and boasts rich online capacities for data retrieval, analysis, and visualization [30]. 

## 3. Various Algorithms Create Possibilities for Customized Analysis

Bioinformatics tools or websites can be used to predict protein structure, to look for conserved domains in a protein, or to annotate genes (Table 4). Data visualization and presentation are an integral part of bioinformatics analysis [31]. The biggest advantages of TBtools are batch processing and the visualization of data, and the interactive graphics generated with TBtools are rich with editable features that provide maximum flexibility for users [32]. Protter supports protein data analysis and protein prediction by visualizing the characteristics of an annotated sequence and associated experimental proteomic data in a protein topological environment. Protter is of great use for comprehensive visualization of membrane proteins and the selection of targeted proteomic peptides [33].

Many tools are also tailored for specific applications (Table 5), including the prediction of transcription factor binding sites and the exploration of large-scale genomic variation data. For example, the PlantPAN database hosts a comprehensive list of transcription factors and their cognate binding sites. TRANSFAC and PlnTFDB are comprehensive databases of plant transcription factors, and AGRIS contains a database of Arabidopsis transcription factors, which can be used to predict transcription factor binding sites in plant promoter regions [34]. SnpHub can be used to retrieve, analyze, and visualize large-scale genomic variation data by specifying samples and lists of specific genomic regions [35].

Outside of dedicated web tools, various algorithms can be used to empower data integration and analysis, such as Python, R, and Perl. Python and R are perhaps more widely used in bioinformatics than Perl. R has powerful statistical functions, which are very useful for processing large experimental datasets, together with a graphics solution for data exploration [36]. Python is better suited for building databases and web applications and is better for developing utilities [37]. While the basic introductory programming paradigm in R relies on so-called functions hosted by user-written packages, Python’s programming paradigm is based on design flow. Although R code might not be as human-readable as Python’s, R is overall better suited to biologists with no strong programming background. Based on these programming languages, various scripts have been developed to efficiently analyze data. For example, R uses a k-means function for clustering analysis and can draw Manhattan plots produced from genome-wide association studies (GWASs) with the qqman package [38].

## 4. Application of Bioinformatics in Plant Breeding

Plant breeding aims to produce new plant varieties. This long-term activity begins with basic research and often takes many years, thus necessitating a significant financial investment [39]. Genomics-assisted breeding is an effective and economical strategy and is thus widely applied in crop breeding. Genomics may help to understand the organization and function of biological systems and has the potential to track the molecular changes during development under different conditions, such as changes in plant physiology, pathogen pressure, or in the environment [40]. Samples for genomics studies can be collected from the same or different individuals from one species or from different species [41]. In addition, comparative genomics allows the study of specific traits in related plants by capitalizing on sequence conservation between species with small genomes (easier to study) and those with large and complex genomes (more difficult to study, but including most current crop species). For example, in Chrysanthemum, GWASs have been used to explore genetic patterns and identify favorable alleles for several ornamental and resistance traits, including plant structural and inflorescence traits, waterlogging tolerance, aphid resistance, and drought tolerance [42]. Su et al. transferred a major SNP co-isolated with waterlogging tolerance in Chrysanthemum to a PCR-based derived cut amplified polymorphism sequence (dCAPS) marker with an accuracy of 78.9%, which was verified in 52 cultivars or progenitors [43]. Chong et al. developed two dCAPS markers associated with the flowering stage and diameter of the head in Chrysanthemum. These dCAPS markers have potential applications in the molecular breeding of Chrysanthemum [44]. These techniques will provide new powerful tools for future Chrysanthemum breeding.

### 4.1. Bioinformatics Can Be Applied to Breed Germplasm with High Yield and Quality

Bioinformatics can be applied to crop breeding to improve yield and quality (Figure 1) [45]. Through a bioinformatics analysis of the genes related to seed germination, seedling growth and reproductive yield, and artificial interference with relevant genes, crops can be further improved [46]. For example, the adaptability, yield, and quality of rapeseed (*Brassica napus*) have been the target of genetic improvement via breeding [47]. In addition, bioinformatics methods can help measure the best leaf angle for the highest photosynthetic rates, to create plants with an optimal leaf angle, which may increase the accumulation of organic matter in plants.

### 4.2. Bioinformatics Can Accurately Predict Plant Growth and Conditions

Plant leaf angle (Figure 2) has a great impact on plant photosynthesis. Reasonable close planting is an effective method to increase crop yield by increasing the photosynthetic area. Leaf angle is a key character of plant structure and a target of crop genetic improvement. Under high density planting, upright leaves can better capture light, which improves photosynthetic efficiency, ventilation, and stress tolerance, and ultimately, increases grain yield. Considerable evidence has shown that auxin, gibberellins (GAs), lactones (SLs), and ethylene contribute to leaf angle formation [48]. For example, LsNRL4 deletion in lettuce resulted in chloroplast enlargement, reduced the amount of cell space allocated to chloroplasts, and caused defective secondary cell wall biosynthesis in leaf joints. Overexpression of LsNRL4 significantly decreased leaf angle and improved photosynthesis [49]. In the applications, it is possible to use the bioinformatics method to measure the stronger leaf angle of plant photosynthesis, from the perspective of bioinformatics analysis to create the optimal angle of leaves, and to increase the accumulation of organic matter in plants. For example, the QTL of the opposite leaf angle in maize and the key part of regulating leaf angle in the leaf tongue region were studied [50], and the leaf angle extractor (LAX) developed based on the image-processing framework of MATLAB, which quantifies corn and sorghum leaf angles from image data. LAX can be used to analyze changes in leaf angle across multiple genotypes and measure their response to drought stress, and it is particularly used in tracking individual plants over time [51].

Mineral malnutrition has significant effects on plant development (Figure 3) [52], especially the lack of nitrogen, potassium, calcium, phosphorus, and iron, which is a huge problem for agriculture [53]. Watching for early warning signs of deficiency in these elements and knowing how to remedy this problem is of great significance for agriculture. Current methods employed to determine nutritional deficiency in plants rely on the analysis of mineral contents in the soil and/or in the plant. However, these methods are expensive and time consuming. The physiological state of legumes changes when plants experience deficiency in any of these macro- and microelements, which can be reflected in the changes of the chlorophyll fluorescence transient [54]. Nutritional deficiency in the above elements is accompanied by damage to the electron transport chains on the donor and acceptor sides of photosystem II (PSII) and PSI [55]. Based on chlorophyll fluorescence data, one study used back propagation within artificial neural networks to identify missing elements [56]. From this analysis, researchers then proposed a new method for determining plant nutrient deficiencies based on rapid chlorophyll fluorescence measurements, which can accurately predict whether legumes lack N, P, K, Ca, and Fe before the plants start to show obvious signs of a deficiency. This new method clearly illustrates the potential for incorporating bioinformatics into the early detection and prevention of plant nutritional deficiency.

### 4.3. Automation for Agriculture

The advancement of automation and digital technologies has led to the development and use of automated robots. The development of smart agriculture has changed the time-consuming and labor-intensive nature of traditional agriculture and has improved the efficiency of agricultural practices [57,58]. For instance, an electric sprayer can be assembled on a robot that can calculate the leaf density of the plants beneath, which is then used by the controller to adjust the air flow, water rate, and the water density of the sprayer for optimal watering and the precise application of pesticides, thus limiting the potential for chemical residues in the soil [59]. An automated irrigation system was later developed that provides both efficient water use and real-time monitoring of the environment. The irrigation system uses a NodeMCU ESP32 microcontroller to collect environmental data such as humidity, temperature, and soil moisture levels through sensors that can irrigate plants at specific times and is also equipped with a passive infrared sensor to detect intruders in the vicinity of the farmland and warn the farmer if severe conditions, such as extreme temperatures, are detected. This system can therefore automatically irrigate farmland without human intervention and allows farmers to monitor and manually control irrigation with the use of a smartphone app [60]. 

Some researchers have applied bioinformatics to breeding and developed a bioinformation breeder, which can transfer the good traits of donor crops to recipient crops after processing, so that the good traits of recipient crops coincide with the good traits of donor crops. A large number of experimental results show that the biological energy breeding machine can realize the transfer of biological information across space, realize the donor plants biofield through a biological information transfer machine across space in a relatively short time, and influence or induce the change of the receptor plant’s genetic traits. This new breeding method operation is simple, low cost, and does not destroy the organism’s own genes. To meet the health needs of human beings, new varieties with high yield, high quality, and harsh environment resistance are produced at the same time, thus creating a new stage of breeding industry [61]. However, this bioinformation breeder is based on a special kind of bioenergy—biomicrowave. Although the magnitude of biomicrowave is much lower than an electron volt, a large number of experiments have shown that this weak energy can not only transmit biological information, but also affect the protein activities of biological receptors across space. However, because the biological microwave (approximately 4–20 μm) is the lowest energy state in nature, which is involved in quantum, biology, electronics, microwave, and many other scientific and technological fields, it has not been developed and widely applied [62].

### 4.4. Accurate Prediction of Experimental Results and Transgenic Phenotypes

In plant research, genotype-phenotype prediction has traditionally used statistical methods. For example, two statistical methods, autoregressive (AR) and Markov chain (MCMC), are used to predict the growth trends of plants by using the Normalized Difference Vegetation Index (NDVI) [63]. Importantly, the application of machine learning to genotype–phenotype prediction will facilitate the study of the roles played by various molecular components in shaping plant phenotypes. The advantages of machine learning over traditional statistical methods are that machine learning can distinguish between different types of genomic regions, and also predict the location of genomic crossovers, which expands the application of machine learning to population genetics [64].

Plant breeders increasingly rely on genomic selection, that is, the selection of favorable alleles at specific loci, which requires the mapping and localization of quantitative trait loci (QTL) to describe the underlying genetic architecture of a given trait and clone the causal alleles. For example, QTLs have been identified in durum wheat for protein grain content [65], high grain yield [66], disease resistance [67,68], and quality traits [69]. In Chrysanthemum, several QTLs controlling flower color, flowering time, ray floret number, and disc floret number were identified by multiallelic QTL analysis. Similarly, seven QTLs affecting tuber shape were detected in the potato (*Solanum tuberosum*) [70]. In maize, several QTLs were described for both insect resistance [71] and for multiple drug resistance, relating to disease resistance research [72]. QTLs were also reported for thrips resistance in pepper (*Capsicum annuum*) [73] and to explore the genetic basis of cooking time and protein concentration of dried beans from *Phaseolus vulgaris* L. [74]. Several researchers have explored machine learning methods for QTL localization (mainly for pre-screening), but their use is still limited. Alternatively, deep learning has been successfully applied to plant phenotype identification. For example, a Convolutional Neural Network (CNN) was used to detect and classify spikes and spikelets in wheat images to study plant development [75].

## 5. Application of Bioinformatics in Research on Plant Disease Resistance and Various Abiotic Stresses

Plants face many environmental challenges, such as diseases and insect pests, light, extreme temperatures, water availability, soil salinity, and other stresses [76]. Plant tolerance to biotic and abiotic stresses is controlled by cross adaptation. Different stresses cause distinct changes in plants (Figure 4). Under high light or drought conditions, perturbations in the Calvin–Benson–Bassham (CBB) cycle generate reactive oxygen species (ROS) and activate chloroplast-to-nucleus retrograde signaling to confer drought tolerance [77]. Under high-temperature stress conditions, the steady-state levels of the nucleocytoplasmic immune regulators ENHANCED DISEASE SUSCEPTIBILITY 1 and PHYTOALEXIN DEFICIENT 4 also decrease [78]. At low temperatures, cold signals lead to an increased accumulation of salicylic acid and pathogenesis-related (PR) proteins [79]. In addition, the transcription factor ELONGATED HYPOCOTYL 5 (HY5) accumulates to promote COLD-REGULATED gene expression and an acclimation to cold [80]. Following its cold-induced monomerization and nuclear translocation, NONEXPRESSER OF PR GENES 1 functions with the heat shock factor HSFA1 to confer freezing tolerance [81]. The cell wall undergoes softening and remodeling under high salinity [82]. The fluidity, integrity, and function of phospholipid membranes are affected by different stresses [83]. Chrysanthemum dwarf viroids (CSVd) can invade the leaf primordium and cells very close to the apical dome or even the outermost layer of the apical dome. The lack of CSVd in shoot tip seriously affected the asexual propagation of chrysanthemum [84]. Some researchers have developed a plant disease resistance protein predictor (RD-RFPDR), which showed to be sensitive and specific in identifying DR proteins after excluding data imbalance. This can provide a method for predicting plant disease resistance proteins [85].

### 5.1. Predicting Plant Resistance Based on Key Indicators

The molecular regulatory networks underlying plant stress resistance and adaptation can be studied in conjunction with genomics and proteomics [86]. Technologies such as transcriptome deep sequencing (RNA-seq) can provide a wealth of information on differential gene expression to study key genes involved in plant stress tolerance. Bioinformatics, next-generation sequencing, and genomics help us to better understand the molecular mechanisms of plant tolerance to different stress conditions and can aid in breeding novel plant varieties and improving crop quality [87].

Several groups used RNA-seq analysis to investigate the transcriptional changes in different crop plants [88]. For example, genes whose transcript levels increased in response to salt and drought stress were identified in chickpea (*Cicer arietinum*) at different developmental stages [89]; differentially expressed genes were described in cold-tolerant and cold-sensitive varieties of sorghum in response to cold stress [90] Likewise, abiotic stress was shown to induce transcriptional reprogramming in poplar (*Populus trichocarpa*) [91]; changes in gene expression were reported in heat-resistant rice [92]; and genes potentially related to heat stress tolerance were proposed in spinach (*Spinacia oleracea*) [93]. Genomics is a powerful tool to study the molecular mechanism behind plant stress tolerance, as recently illustrated by the studies of salt stress in plants [94] and of drought tolerance in rice [95]. In addition, a comparative analysis of transcriptome datasets in tobacco revealed the genes of resistance and susceptibility to *Phytophthora nicotiana*, which provided valuable resources for breeding resistant tobacco plants [96]. As for the study on sorghum, differential gene expression analysis identified more than 3000 specific genes in two sorghum cultivars with resistance and susceptibility to anthracnose disease, and showed significant changes in the expression of these genes after inoculation with anthracnose [97].

### 5.2. Genome Annotation Is Available and Feasible

Identifying genes from the mass of genomic sequences is an important task in bioinformatics. The annotation of genome sequences can encounter one of two scenarios: (1) gene annotation is performed for a few target sequences with the aim of understanding the possible complement of functional genes, for example, the members of a specific gene family; and (2) gene annotation is performed at the whole-genome level for a newly sequenced genome [98]. In the first scenario, gene annotation of the target sequences can be performed using free online gene prediction platforms and search platforms; the second scenario usually calls upon ab initio prediction algorithms.

Bioinformatics is instrumental for many experiments that explore how plants respond to adverse growth conditions and adapt accordingly. Eelgrass (*Zostera marina*), for example, lacks stomata because the key genes responsible for stomatal developmental are missing in this species, perhaps as an adaptation to life in the ocean. Variation in the CAZyme protein family resulted in a thickening of the cuticle of eelgrass. Changes in sucrose synthase and transport genes also led to changes in the metabolic pathways [99]. These studies were only made possible by bioinformatics approaches. 

To adapt to a high salinity environment, halophytes have evolved many unique characteristics. Bioinformatics has also greatly facilitated the comparative analysis of halophytes to reveal their adaptations to saline-alkali soils [100]. Selaginella is a kind of xerophyte that has adapted to arid environments by thickening the cuticles of leaves, the causes of which were analyzed using bioinformatics [101]. Carnivorous plants use animals and insects as their source of nutrition; bioinformatics revealed the molecular basis that led to the evolution of carnivorous plants [102]. With more genomes available, bioinformatics also uncovered the evolutionary lineages of plants. For example, in the study of water lilies (*Nymphaea colorata*), a phylogenetic analysis determined that Amborellales and Nymphaeales are the successive sister lineages to all other extant angiosperms [103].

### 5.3. Multiple Genes Can Be Merged to Analyze Their Roles in Various Resistance

Ab initio methods are an important research area in bioinformatics, and many prediction algorithms and corresponding procedures have been proposed and applied successively. Unlike homology-based comparison methods, ab initio prediction methods are based on the statistical characteristics of coding regions and gene signals for the prediction of gene structure [104].

The transfer of exogenous DNA into plant genomes has greatly promoted the progress of basic and applied plant research [105]. Plant genome engineering can be harnessed to alter plant metabolism and produce the desired metabolites, as well as improving crop traits. Different transformation methods and strategies have been developed to allow the simultaneous production of multiple plant- or non-plant-derived recombinant proteins in transgenic plant hosts [106]. Future studies on improving plant stress resistance should focus more on the combination of multiple approaches, such as the introduction of multiple genes simultaneously into transgenic plants. An alternative to stacking multiple genes in transgenic plants is the use of iterative or serial transformation strategies, whereby the genes of interest are introduced one at a time through successive rounds of transformation [107,108] or by sexual crossing of transgenic lines, carrying different transgenes to bring them together in the same background [109,110,111,112]. For example, genes involved in osmoprotectant biosynthesis have been co-expressed with other stress resistance-related genes, such as ion transporters and transcription factors.

## 6. Perspective

In the era of big data, bioinformatics faces opportunities and challenges for its application to agriculture. Learning and developing more bioinformatics tools will help integrate all existing bioinformation resources and provide support for effective breeding and plant resistance analysis [113].

Food production systems are under tremendous pressure due to the continuous growth of the human population. Many of the world’s ecosystems are already overexploited, and meeting the growing demand for food by expanding arable land use is not possible [114]. Indeed, according to the Food and Agriculture Organization (FAO), only 10% of future growth in agricultural production will come from the expansion of acreage, while the remaining 90% must come from yield hikes [115]. The development of genomics technology has provided huge technical support for breeders, who have been able to continuously breed new varieties that are more adaptable to the environment and have higher yields, leading to the continuous improvement of seed replacement rate.

The bioinformatics era initiated by NGS has revolutionized the design of experiments in molecular biology, substantially contributing to the growth of scientific knowledge while influencing relevant applications in many different aspects of agriculture [116]. Data from different research areas support the co-development and advancement of molecular knowledge through extensive efforts, with bioinformatics being the driving force. Organization, detection, integration of data, and data sharing are facilitating multidisciplinary interactions, expanding resources, and disseminating common methods [117]. Bioinformatics is thus revolutionizing agricultural practices and production, providing knowledge and tools to improve product quality, and improving strategies to counteract environmental stresses, diseases, and parasites [8,118]. Bioinformatics is evolving, and we have great hopes for the integration of bioinformatics in plant research.

## Figures and Tables

**Figure 1 plants-11-03118-f001:**
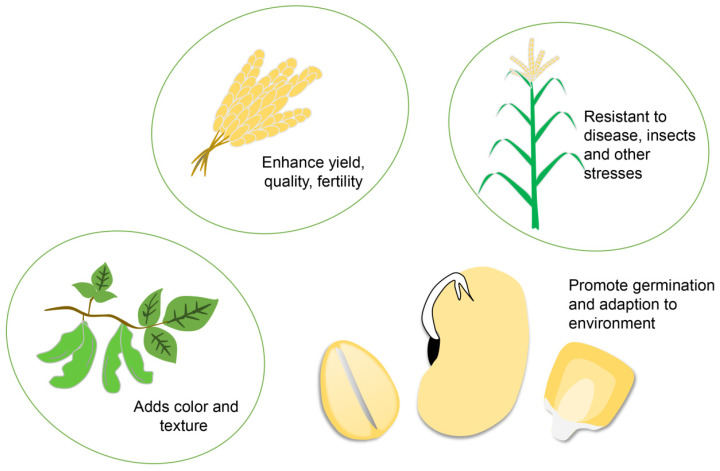
Breeding indicators that can be improved using bioinformatics. Breeding aims to integrate various indicators such as yield, quality, fertility, disease resistance, insect resistance, collapse resistance, as well as salt resistance and adaptability to adverse environments such as drought, waterlogging, high temperature, low temperature, and salinity to achieve superior varieties.

**Figure 2 plants-11-03118-f002:**
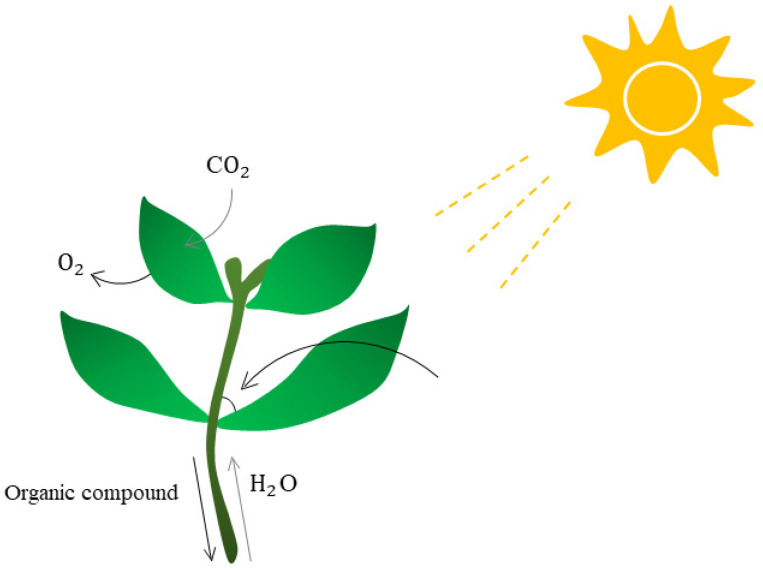
The effect of plant leaf angle. The angle of plant leaves has a strong influence on many plant activities. When the plant leaf angle is optimal, plants enjoy a high photosynthetic rate, thus producing more organic matter. Creating new species with optimal leaf angles could help plants synthesize more organic matter, leading to better growth and, in turn, increased survival and yield.

**Figure 3 plants-11-03118-f003:**
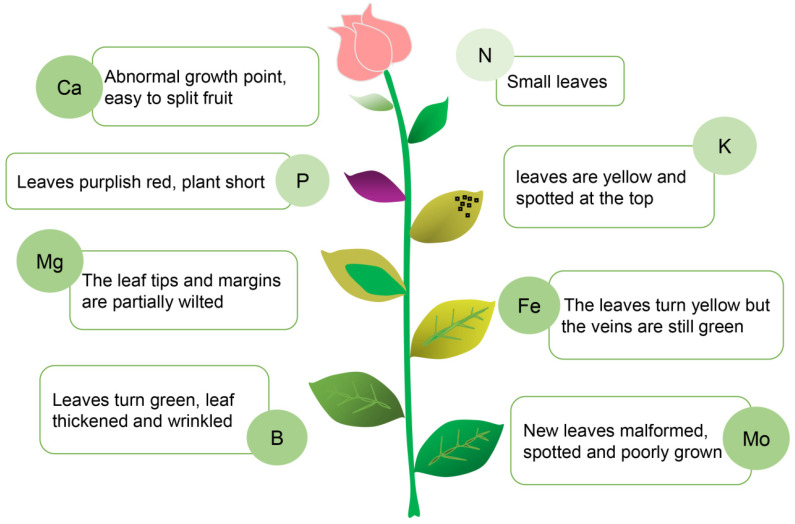
The effects of specific mineral nutrient deficiency on plants. The nutritional status of a plant is reflected in leaf shape and leaf color. Based on the information collected by visualizing the leaves, the appropriate nutrients can be supplied to the plant in a targeted manner to achieve precise fertilization. This chart is an example of a moonflower.

**Figure 4 plants-11-03118-f004:**
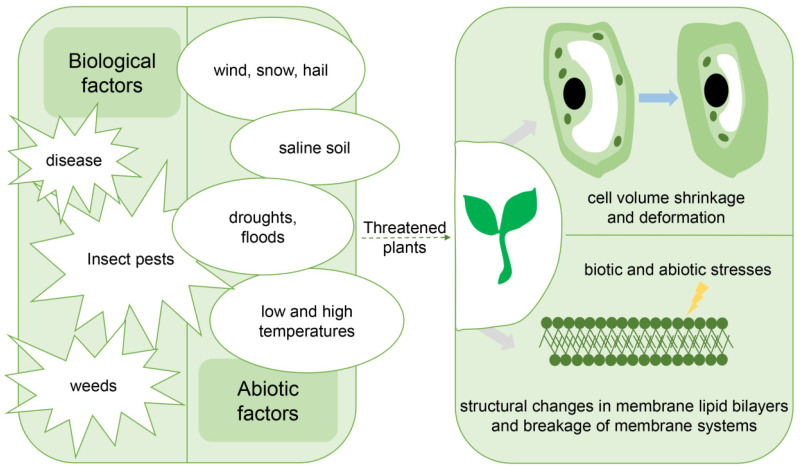
Some plant resistance factors and their effects. Plants are continuously exposed to biotic and abiotic stresses. Plant tolerance to biotic and abiotic stresses is controlled by cross tolerance. Although each stress will impose distinct physical and chemical effects on plants, most stresses will often directly or indirectly lead to water stress at the whole-plant scale, causing water loss, cell volume shrinkage and deformation, structural changes in the membrane lipid bilayer, and degradation of the membrane system. Therefore, a common basis for plant resistance is osmoregulation, changes in the interaction between membrane lipids and membrane proteins, and repair activities.

**Table 1 plants-11-03118-t001:** General databases used for data integration and presentation.

URL	Note	Description
http://bigd.big.ac.cn/databasecommons/ Accessed on 4 May 2022.	Comprehensive publicly available data repository covering a wide range of organisms	Bigd database consolidates all the information collected about the database. Each database is classified by data type, category, subject, and location, so that people can easily find a specific collection of databases of interest.
https://www.expasy.org/ Accessed on 4 May 2022.	Covers a wide range of biological research databases and software tools	Expasy database is divided into several areas: DNA, RNA, protein, population, cell, etc. According to omics, it is divided into proteome, genome, transcriptome, structure analysis, population genetics, and so on.
https://www.agbiodata.org/ Accessed on 4 May 2022.	Integrated platform of agricultural biological databases and related resources	It is a consortium of agricultural biological databases that integrate standards and best practices for the acquisition, display, and reuse of genomic, genetic, and breeding data.
https://phytozome.jgi.doe.gov/pz/portal.html Accessed on 4 May 2022.	Plant Comparative Genomics Repository	Phytozome database, the Plant Comparative Genomics portal of the Department of Energy’s Joint Genome Institute, provides a hub for accessing, visualizing, and analyzing JGI-sequenced plant genomes, as well as selected genomes and datasets sequenced elsewhere.
http://harvest.ucr.edu/ Accessed on 4 May 2022.	Platform for Crop EST sequences and related molecular information	HarvEST database includes various functions, such as microarray content design, SNP identification, genotyping platform design, comparative genomics, and the coupling of physical and genetic profiles.
https://www.uniprot.org/ Accessed on 4 May 2022.	Protein sequence and functional information resource database and analysis platform	Uniprot database is the world’s leading resource for high-quality, comprehensive, and freely accessible protein sequence and functional information.
http://www.plantgdb.org/ Accessed on 4 May 2022.	Plant Genome Sequence Database	Plantgdb database includes software, visualization, and data access portals that implement novel prediction algorithms, as well as a network infrastructure environment implementation of development tools for distributed computing, protocol sharing, and analysis of source records.
https://mpss.danforthcenter.org/index.php Accessed on 4 May 2022.	NGS database, including small RNAs and genome resources for plants	Meyers Lab database focuses on many aspects of plant small RNAs, including their major roles in gene and transposable regulation, but also their biogenesis and evolution. Includes small RNA sequencing, cut target RNA sequencing, and a variety of informatics tools.
http://metacrop.ipk-gatersleben.de Accessed on 4 May 2022.	Crop Metabolism Pathway Database	Metacrop database summarizes various information about metabolic pathways in crops and allows the automatic export of information to create detailed metabolic models.

**Table 2 plants-11-03118-t002:** Databases specific for Arabidopsis.

URL	Note
http://www.arabidopsis.org Accessed on 4 May 2022.	The most commonly used repository of Arabidopsis genetic and molecular biology data
http://rarge.gsc.riken.jp/ Accessed on 4 May 2022.	Arabidopsis cDNA, mutant, and microarray database
http://www.athamap.de/ Accessed on 4 May 2022.	A genome-wide database of putative transcription factor binding sites in Arabidopsis
http://www.plprot.ethz.ch/ Accessed on 4 May 2022.	Arabidopsis plastid protein database
http://seedgenes.org/ Accessed on 4 May 2022.	Database of key Arabidopsis developmental genes
http://suba.live/ Accessed on 4 May 2022.	Subcellular localization database for Arabidopsis proteins
http://atrm.cbi.pku.edu.cn/ Accessed on 4 May 2022.	Arabidopsis transcriptional regulatory mapping database
http://wanglab.sippe.ac.cn/rootatlas/ Accessed on 4 May 2022.	Arabidopsis root single-cell RNA-seq database
http://ipf.sustech.edu.cn/pub/athrna/ Accessed on 4 May 2022.	Arabidopsis RNA-seq data resources
http://signal.salk.edu/ Accessed on 4 May 2022.	A database showing all T-DNA insertions and methyl group data

**Table 3 plants-11-03118-t003:** Databases used for major crops.

URL	Note
http://www.ricedata.cn/index.htm Accessed on 4 May 2022.	National Rice Data Center
http://signal.salk.edu/cgi-bin/RiceGE Accessed on 4 May 2022.	Rice functional genome expression database
https://shigen.nig.ac.jp/rice/oryzabase/ Accessed on 4 May 2022.	Rice genetics and genomics database
http://www.wheatgenome.org/ Accessed on 4 May 2022.	Wheat genome information database
http://earth.nig.ac.jp/~dclust/cgi-bin/index.cgi Accessed on 4 May 2022.	Barley germplasm resources and genome analysis database
http://maize.jcvi.org/ Accessed on 4 May 2022.	Maize genome database
https://www.maizegdb.org/ Accessed on 4 May 2022.	Maize genome and genetic analysis platform
https://soybase.org/ Accessed on 4 May 2022.	Soybean genomics and molecular biology database
http://www.ildis.org/LegumeWeb/ Accessed on 4 May 2022.	International legume database and information service
https://www.cottongen.org/ Accessed on 4 May 2022.	Cotton genomics, genetics, and breeding database
http://ted.bti.cornell.edu/ Accessed on 4 May 2022.	Tomato functional genome database
http://ted.bti.cornell.edu/epigenome/ Accessed on 4 May 2022.	Tomato epigenome database
http://tea.solgenomics.net/ Accessed on 4 May 2022.	High-resolution mapping and search tool for tomato genes and their products
http://tomexpress.toulouse.inra.fr/ Accessed on 4 May 2022.	Tomato transcriptome data visualization and analysis platform
https://solgenomics.net/ Accessed on 4 May 2022.	Genome sequencing database of Solanaceae species
http://gabipd.org/projects/Pomamo/ Accessed on 4 May 2022.	Potato bioinformatics database

**Table 4 plants-11-03118-t004:** Bioinformatics tools and websites that can be used in plant research.

Database Name	URL	Note
Tbtools	https://github.com/srbehera11/stag-cns Accessed on 4 May 2022.	An integrated toolkit for interactive analysis of big biological data
SMART	http://smart.embl-heidelberg.de/ Accessed on 4 May 2022.	Protein conserved domain prediction tool
STAG-CNS	https://github.com/srbehera11/stag-cns Accessed on 4 May 2022.	A sequentially conserved non-coding sequence discovery tool for an arbitrary number of species
FED	http://www.hi-tom.net/FED Accessed on 4 May 2022.	Genome editing exogenous component detection platform
MAFFT	https://mafft.cbrc.jp/alignment/server/ Accessed on 4 May 2022.	Online sequence matching tool
Protter	http://wlab.ethz.ch/protter/start/ Accessed on 4 May 2022.	Online protein structure mapping tool
EvolView	https://www.evolgenius.info/evolview/ Accessed on 4 May 2022.	Web-based tools for visualizing, annotating, and managing system trees
iTOL	https://itol.embl.de/ Accessed on 4 May 2022.	Online tool for displaying, annotating, and managing system development trees

**Table 5 plants-11-03118-t005:** Bioinformatics tools specific to applications in plant research.

Database Name	URL	NOTE
BAR	http://www.bar.utoronto.ca/welcome.htm Accessed on 4 May 2022.	Plant biology analysis tools platform
CRISPR-P	http://crispr.hzau.edu.cn/CRISPR2/ Accessed on 4 May 2022.	Improved CRISPR/Cas9 toolkit for plant genome editing
ACT	https://www.michalopoulos.net/act/ Accessed on 4 May 2022.	Arabidopsis co-expression analysis tool
OryGenesDB	http://orygenesdb.cirad.fr/ Accessed on 4 May 2022.	An interactive tool for reverse genetics studies in rice
T-DNA Express	http://signal.salk.edu/cgi-bin/tdnaexpress Accessed on 4 May 2022.	Arabidopsis gene targeting tool
Plant MetGenMAP	http://bioinfo.bti.cornell.edu/cgi-bin/MetGenMAP/home.cgi Accessed on 4 May 2022.	Web-based tools for comprehensive mining and integration of gene expression and metabolite changes in the context of biochemical pathways
iTAK	http://itak.feilab.net/cgi-bin/itak/index.cgi Accessed on 4 May 2022.	Software packages for identifying and classifying plant transcription factors and protein kinases
PlantPAN	http://plantpan2.itps.ncku.edu.tw/ Accessed on 4 May 2022.	Tools for detecting transcription factor binding sites in plants
SnpHub	http://guoweilong.github.io/SnpHub/ Accessed on 4 May 2022.	A unified web server framework for exploring large-scale genomic variation data

## Data Availability

Data sharing not applicable to this article as no datasets were generated or analysed during the current study.

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
