# Peer review of "Bioinformatics in Plant Breeding and Research on Disease Resistance"

_plants, 2022, doi:10.3390/plants11223118_

Round 1
Reviewer 1 Report
The English language must be reviewed throughout the manuscript. As concern the chosen title, the application of bioinformatics to plant breeding is quite well developed, but very little is said about resistance of plants to pathologies. Therefore, the organization of the entire manuscript and its development should be rethought and / or revised. Furthermore, also the aspect of plant breeding itself is not very well defined. It is necessary to develop and articulate better paragraphs 4 and 5, otherwise the review risks to be a mere list of databases and algorithms used for the study of plant physiology. For these reasons, the manuscript cannot be accepted for publication in “Plants” in this form but needs major revisions.
In the Introduction section, first the authors declare that they want to provide a description of bioinformatics tools and their application, focusing on plant breeding and research on disease resistance (biotic stress, ll. 29-30), and then they deal with plant morphophysiological, genetic studies and works on molecular mechanisms underlying plant responses to abiotic stress. What about disease resistance and biotic stress? In the Introduction, a more in-depth excursus should be made on what has already been done in bioinformatics and plant breeding and resistance to diseases.
You could use, for example, some of the following works on the topics:
Vassilev, D., Leunissen, J., Atanassov, A., Nenov, A., & Dimov, G. (2005). Application of bioinformatics in plant breeding. Biotechnology & Biotechnological Equipment, 19(sup3), 139-152.
Blätke, M. A., Szymanski, J. J., Gladilin, E., Scholz, U., & Beier, S. (2021). Advances in Applied Bioinformatics in Crops. Frontiers in Plant Science, 12, 640394.
Ujjawal Kumar Singh Kushwaha. (2017). Role of Bioinformatics in Crop Improvement. Global Journal of Science Frontier Research, 17(1), 13–23.
Mao Y, Sun X, Shen J, Gao F, Qiu G, Wang T, Nie X, Zhang W, Gao Y and Bai Y (2019) Molecular Evolutionary Analysis of Potato Virus Y Infecting Potato Based on the VPg Gene. Front. Microbiol. 10:1708. doi: 10.3389/fmicb.2019.01708.
Alemu, K. (2015). The role and application of bioinformatics in plant disease management. Advances in Life Science and Technology, 28, 28-33.
Khan, A., Singh, S., Singh, V.K. (2021). Bioinformatics in Plant Pathology. In: Singh, K.P., Jahagirdar, S., Sarma, B.K. (eds) Emerging Trends in Plant Pathology . Springer, Singapore. https://doi.org/10.1007/978-981-15-6275-4_32.
Ll. 95-97. “For example, The Arabidopsis Information Resource (TAIR) database allows users to download gene sequences in bulk, while the SeqViewer tool in TAIR provides a simple too to visualize genes.” …..while the SeqViewer in TAIR also provides a simple tool to visualize genes
Also beware of spelling errors; for example:
Ll. 152-154: Plant breeding aim to produce new plant varieties with improved traits or new traits, including high yield and quality……. Use: Plant breeding aims to produce new plant varieties……
L. 165, title: 4.1. Bioinformatics can be applied at to breed germplasm with high yield and quality
Use
4.1. Bioinformatics can be applied to breed germplasm with high yield and quality
Or
4.1. Bioinformatics can be applied at breeding germplasm with high yield and quality
Ll. 175-178: Plant leaf Angle angle (Figure 2) has a great impact on plant photosynthesis, in the application can use bioinformatics method to measure the strongest leaf Angle angle of plant photosynthesis, from the perspective of bioinformatics analysis to create the optimal Angle angle of leaves, increase the accumulation of organic matter in plants.
Why did you use the expression leaf “Angle angle” to indicate the leaf angle or lamina inclination?
Why repeat the word “angle” twice? I can not understand.
Moreover, please, please enter some references for this interesting topic, which has been studied and treated in scientific publications since the late 1980s.
L. 214: Bioinformatics applied to breeding is mainly a bioinformatics breeding machine.
The concept that you want to express with this sentence is not very clear. Please explain better.
L. 219:….the organism’s own genes, To meet the…..Full stop after genes?
Ll. 222-225. This last sentence of the paragraph is not clear. Please explain better, perhaps with the help of some references.
Paragraph 5 (Application of bioinformatics in research on plant disease resistance).
Ll. 252-267. Again, here the authors want to address the issue of plant resistance to disease, but they do too much talk about abiotic stresses, high light, high- and low-temperature, cold, salinity. Why?
Paragraph 5.1
Instead of or in addition to use the reference 67 on human microbiome, now it is also available research on plant microbiome, e. g.:
Lucaciu R, Pelikan C, Gerner SM, Zioutis C, Köstlbacher S, Marx H, Herbold CW, Schmidt H and Rattei T (2019). A Bioinformatics Guide to Plant Microbiome Analysis. Front. Plant Sci. 10:1313. doi: 10.3389/fpls.2019.01313
Knief C. Analysis of plant microbe interactions in the era of next generation sequencing technologies. Front Plant Sci. 2014 May 21;5: 216. doi: 10.3389/fpls.2014.00216. PMID: 24904612; PMCID: PMC4033234.
Ll. 276-286. Again, there are a lot of research activities on plant genomics and disease resistance; e. g.:
Meng, H., Sun, M., Jiang, Z. et al. Comparative transcriptome analysis reveals resistant and susceptible genes in tobacco cultivars in response to infection by Phytophthora nicotianae. Sci Rep 11, 809 (2021). https://doi.org/10.1038/s41598-020-80280-7
Natarajan P, Ahn E, Reddy UK, Perumal R, Prom LK and Magill C (2021) RNA-Sequencing in
Resistant (QL3) and Susceptible (Theis) Sorghum Cultivars Inoculated With Johnsongrass Isolates of Colletotrichum sublineola.Front. Genet. 12:722519. doi: 10.3389/fgene.2021.722519
Throughout paragraph 5, the reported examples of plant resistance prediction and genomic annotation concern plant abiotic stresses and not the plant resistance to diseases. If the purpose of the review is to focus the use of bioinformatics in research on the response of plants to disease, it is necessary to report cases of use of this powerful tool in the study of plant pathology.
Please integrate these parts or modify them accordingly. In fig. 4 there is an attempt to list abiotic and biotic stresses affecting plants. If at the genetic level the plant response to disease is controlled by genes common to both abiotic and biotic stress-response (cross-tolerance), it must be explained in the text and documented.
Insert more details in the description of Table 1.
References
Attention!
All the citations of the articles in the References section are missing the name of the journal in which they are published and, the chapters of books cited lack the editorial indication of the text. Possible? Please, check.
It is rather tiring to follow the reading of the review without any indication of the bibliographic origin of the references used.
Please, check also the editing of references; for example:
Ref. 26: None %J Africa Research Bulletin: Economic…. None %?
Author Response
- The English language must be reviewed throughout the manuscript. As concern the chosen title, the application of bioinformatics to plant breeding is quite well developed, but very little is said about resistance of plants to pathologies. Therefore, the organization of the entire manuscript and its development should be rethought and / or revised. Furthermore, also the aspect of plant breeding itself is not very well defined. It is necessary to develop and articulate better paragraphs 4 and 5, otherwise the review risks to be a mere list of databases and algorithms used for the study of plant physiology. For these reasons, the manuscript cannot be accepted for publication in “Plants” in this form but needs major revisions.
Response: Thank you very much for these suggestions. Yes, we fully agree with you. These advices are constructive and helpful to improve our manuscript. We carefully revised the manuscript accordingly. The main revisions are marked in blue in the revised version. We have also amended some details and have improved the manuscript according to all your comments. The detailed responses are listed one by one below.
- In the Introduction section, first the authors declare that they want to provide a description of bioinformatics tools and their application, focusing on plant breeding and research on disease resistance (biotic stress, ll. 29-30), and then they deal with plant morphophysiological, genetic studies and works on molecular mechanisms underlying plant responses to abiotic stress. What about disease resistance and biotic stress? In the Introduction, a more in-depth excursus should be made on what has already been done in bioinformatics and plant breeding and resistance to diseases.
You could use, for example, some of the following works on the topics:
Vassilev, D., Leunissen, J., Atanassov, A., Nenov, A., & Dimov, G. (2005). Application of bioinformatics in plant breeding. Biotechnology & Biotechnological Equipment, 19(sup3), 139-152.
Blätke, M. A., Szymanski, J. J., Gladilin, E., Scholz, U., & Beier, S. (2021). Advances in Applied Bioinformatics in Crops. Frontiers in Plant Science, 12, 640394.
Ujjawal Kumar Singh Kushwaha. (2017). Role of Bioinformatics in Crop Improvement. Global Journal of Science Frontier Research, 17(1), 13–23.
Mao Y, Sun X, Shen J, Gao F, Qiu G, Wang T, Nie X, Zhang W, Gao Y and Bai Y (2019) Molecular Evolutionary Analysis of Potato Virus Y Infecting Potato Based on the VPg Gene. Front. Microbiol. 10:1708. doi: 10.3389/fmicb.2019.01708.
Alemu, K. (2015). The role and application of bioinformatics in plant disease management. Advances in Life Science and Technology, 28, 28-33.
Khan, A., Singh, S., Singh, V.K. (2021). Bioinformatics in Plant Pathology. In: Singh, K.P., Jahagirdar, S., Sarma, B.K. (eds) Emerging Trends in Plant Pathology . Springer, Singapore. https://doi.org/10.1007/978-981-15-6275-4_32.
Response: Good suggestions and thanks for the advice. We have given an in-depth excursus about the progresses in bioinformatics and plant breeding and resistance to disease.
For example, based on the VPg gene sequence of a PVY (Y virus) isolated from potato, combined with all published sequences in GenBank, it can be inferred of the rate of evolution of PVY and the time to reach the most recent common ancestor using a Bayesian system dynamics framework to advance disease resistance studies in potato [3-5]. Given that multifactorial traits involved in resistance and quality are extremely difficult to be improved, especially in combinations, and some of the genomes of major forage crops such as maize, rice, wheat, sorghum and barley, and the forage legumes soybean and alfalfa, are too large to be analyzed using whole-genome sequencing, attentions have been focused on comparative genomic approaches in order to auxiliary produce seeds with desirable shapes [6-8]. The details are shown in lines 30-40.
In addition, bioinformatics has been applied in plant pathology, such as identifying and predicting "effector" proteins produced by plant pathogens in order to manipulate their host plants. Functional annotation of this pathogen's ability to predict virulence is a critical step in translating sequence data into potential applications in plant pathology [22]. A bioinformatics framework has been proposed to enable stakeholders to make more informed decisions. In this way, a shared biosecurity infrastructure can be established to cater for sustainable global food and fibre production in the context of global climate change and increased chances of accidental disease invasions in the global plant trade [23]. To develop new strategies for plant disease control, researchers must elucidate the complex molecular mechanisms underlying pathogen infection. Whole genome sequencing technology has enabled the sequencing of an increasing number of pathogens and the accumulation of large amounts of genetic data. Therefore, bioinformatics tools for analyzing pathogen genomes, effectors, and interspecific interactions have been developed to understand disease infection mechanisms and pathogenic targets, which will contribute to plant pathology [24]. The details are shown in lines 72-86.
1) List the applications of bioinformatics in plant research; 2) To clarify the application of bioinformatics in plant breeding; 3) To emphasize the advances made by bioinformatics in the study of plant tolerance and disease resistance; 4) The prediction of plant growth by bioinformatics; 5) Calls for greater use of bioinformatic methods in plant research. The details are shown in lines 88-92.
- Ll. 95-97. “For example, The Arabidopsis Information Resource (TAIR) database allows users to download gene sequences in bulk, while the SeqViewer tool in TAIR provides a simple too to visualize genes.” …..while the SeqViewer in TAIR also provides a simple tool to visualize genes
Response: Done. We have reorganized the section shown on lines 123-124.
4.Also beware of spelling errors; for example:
Ll. 152-154: Plant breeding aim to produce new plant varieties with improved traits or new traits, including high yield and quality……. Use: Plant breeding aims to produce new plant varieties……
Response: Done. See line 174 for details.
- L. 165, title: 4.1. Bioinformatics can be applied at to breed germplasm with high yield and quality Use
4.1. Bioinformatics can be applied to breed germplasm with high yield and quality
Or
4.1. Bioinformatics can be applied at breeding germplasm with high yield and quality
Response: Thank you for your suggestion and we have applied the first heading in your suggestion. 4.1. Bioinformatics can be applied to breed germplasm with high yield and quality. The details are shown on line 196.
- Ll. 175-178: Plant leaf Angle angle (Figure 2) has a great impact on plant photosynthesis, in the application can use bioinformatics method to measure the strongest leaf Angle angle of plant photosynthesis, from the perspective of bioinformatics analysis to create the optimal Angle angle of leaves, increase the accumulation of organic matter in plants.
Why did you use the expression leaf “Angle angle” to indicate the leaf angle or lamina inclination?
Why repeat the word “angle” twice? I can not understand.
Moreover, please, please enter some references for this interesting topic, which has been studied and treated in scientific publications since the late 1980s.
Response: Sorry for the inappropriate statement. We have revised as “leaf angle” in the revised version. In addition, we have supplemented the related literatures to insist this opinion shown in lines 206-224 for details.
Reasonable close planting is an effective method to increase crop yield by increasing photosynthetic area. Leaf Angle is a key character of plant structure and a target of crop genetic improvement. Under high density planting, upright leaves can better capture light, which improves photosynthetic efficiency, ventilation and stress tolerance, and ultimately increases grain yield. Considerable evidence has shown that auxin, gibberellins (GAs), lactones (SLs) and ethylene contribute to leaf Angle formation [48]. For example, LsNRL4 deletion in lettuce resulted in chloroplast enlargement, reduced amount of cell space allocated to chloroplasts, and defective secondary cell wall biosynthesis in leaf joints. Overexpression of LsNRL4 significantly decreased leaf Angle and improved photosynthesis [49].
For example, the QTL of opposite leaf Angle in maize and the key part of regulating leaf Angle in the leaf tongue region were studied [50], and the leaf Angle extractor (LAX) developed based on the image processing framework of MATLAB, which quantifies corn and sorghum leaf angles from image data. LAX can be used to analyze changes in leaf Angle across multiple genotypes and measure their response to drought stress, and is particularly used in tracking individual plants over time [51].
- L. 214: Bioinformatics applied to breeding is mainly a bioinformatics breeding machine.
The concept that you want to express with this sentence is not very clear. Please explain better.
Response: Done. Some researchers have applied bioinformatics to breeding and developed a bioinformation breeder, which can transfer the good traits of donor crops to recipient crops after processing, so that the good traits of recipient crops coincide with the good traits of donor crops. We have made relevant modifications to this part. See lines 260-263 for details.
- L. 219:….the organism’s own genes, To meet the…..Full stop after genes?
Response: Done. We have corrected this sentence shown in lines 267-268 for details.
- Ll. 222-225. This last sentence of the paragraph is not clear. Please explain better, perhaps with the help of some references.
Response: Done. Biological microwaves have been explained and references have been added.
However, this bioinformation breeder is based on a special kind of bioenergy -- biomicrowave. Although the magnitude of biomicrowave is much lower than electron volt, a large number of experiments have shown that this weak energy can not only transmit biological information, but also affect the protein activities of biological receptors across space. However, because biological microwave (about 4-20 μm) is the lowest energy state in nature, which is involved in quantum, biology, electronics, microwave and many other scientific and technological fields, it has not been developed and widely applied [62]. The details are shown in lines 270-277.
10.Paragraph 5 (Application of bioinformatics in research on plant disease resistance).
Ll. 252-267. Again, here the authors want to address the issue of plant resistance to disease, but they do too much talk about abiotic stresses, high light, high- and low-temperature, cold, salinity. Why?
Response: Thank you for your advice. Some disease resistance studies in Chrysanthemum and bioinformatic methods for predicting plant disease resistance proteins have been added to the fifth paragraph, which makes up for the lack of content on plant disease resistance in the fifth paragraph. The details can be found in lines 323-329.
Chrysanthemum dwarf viroids (CSVd) can invade the leaf primordium and cells very close to the apical dome or even the outermost layer of the apical dome. The lack of CSVd in shoot tip seriously affected the asexual propagation of chrysanthemum [84]. Some researchers have developed a plant disease resistance protein predictor (RD-RFPDR), which showed to be sensitive and specific in identifying DR proteins after excluding data imbalance. This can provide a method for predicting plant disease resistance proteins [85].
- Paragraph 5.1
Instead of or in addition to use the reference 67 on human microbiome, now it is also available research on plant microbiome, e. g.:
Lucaciu R, Pelikan C, Gerner SM, Zioutis C, Köstlbacher S, Marx H, Herbold CW, Schmidt H and Rattei T (2019). A Bioinformatics Guide to Plant Microbiome Analysis. Front. Plant Sci. 10:1313. doi: 10.3389/fpls.2019.01313
Knief C. Analysis of plant microbe interactions in the era of next generation sequencing technologies. Front Plant Sci. 2014 May 21;5: 216. doi: 10.3389/fpls.2014.00216. PMID: 24904612; PMCID: PMC4033234.
Response: Done. The original reference 67 has been modified to a study of the plant microbiome, as detailed in reference 23 and reference 87.
- Ll. 276-286. Again, there are a lot of research activities on plant genomics and disease resistance; e. g.:
Meng, H., Sun, M., Jiang, Z. et al. Comparative transcriptome analysis reveals resistant and susceptible genes in tobacco cultivars in response to infection by Phytophthora nicotianae. Sci Rep 11, 809 (2021). https://doi.org/10.1038/s41598-020-80280-7
Natarajan P, Ahn E, Reddy UK, Perumal R, Prom LK and Magill C (2021) RNA-Sequencing in
Resistant (QL3) and Susceptible (Theis) Sorghum Cultivars Inoculated With Johnsongrass Isolates of Colletotrichum sublineola.Front. Genet. 12:722519. doi: 10.3389/fgene.2021.722519
Response: Done. Thanks for your suggestions. We have revised the relevant content and added the related content and literature on plant genomics and disease resistance. The details are shown in lines 348-354.
In addition, comparative analysis of transcriptome datasets in tobacco revealed the genes of resistance and susceptibility to Phytophthora nicotiana, which provided valuable resources for breeding resistant tobacco plants [96]. As for the study on sorghum, differential gene expression analysis identified more than 3,000 specific genes in two sorghum cultivars with resistance and susceptibility to anthracnose disease , and showed significant changes in the expression of these genes after inoculation with anthracnose [97].
- Throughout paragraph 5, the reported examples of plant resistance prediction and genomic annotation concern plant abiotic stresses and not the plant resistance to diseases. If the purpose of the review is to focus the use of bioinformatics in research on the response of plants to disease, it is necessary to report cases of use of this powerful tool in the study of plant pathology.
Response: Sorry for the omission. According to your suggestions, the fifth paragraph has been modified accordingly. We have tried our best to make global statements of diseases resistance, but bioinformatics research on plant pathology is relatively few, while research on plant stress resistance can provide the possible direction in future study of plant diseases.
14.Please integrate these parts or modify them accordingly. In fig. 4 there is an attempt to list abiotic and biotic stresses affecting plants. If at the genetic level the plant response to disease is controlled by genes common to both abiotic and biotic stress-response (cross-tolerance), it must be explained in the text and documented.
Response: Thank you for your suggestion. The legend has been modified accordingly according to the content in Figure 4. Plant tolerance to biotic and abiotic stresses is controlled by cross tolerance shown in lines 308-309.
15.Insert more details in the description of Table 1.
Response: Done. Thanks for your suggestion. Table 1 has been improved.
- References
Attention!
All the citations of the articles in the References section are missing the name of the journal in which they are published and, the chapters of books cited lack the editorial indication of the text. Possible? Please, check.
It is rather tiring to follow the reading of the review without any indication of the bibliographic origin of the references used.
Please, check also the editing of references; for example:
Ref. 26: None %J Africa Research Bulletin: Economic…. None %?
Response: Thank you for your comments. Sorry for the mistakes. The references have been sorted out and modified.
Once again, thank you very much for your professional comments and suggestions. We earnestly appreciate your professional work and we hope that this revision will be deemed suitable for publication in Plants.
Reviewer 2 Report
1. Check that the positions of the first and second cited articles in the citation list have not changed.
2. The reference N 37 is an Inappropriate reference. This book makes no mention of the method you describe. See this articles: "Identification of nutrient deficiency in plants by artificial intelligence."
3. Missing authors in reference N 46
4. Check the URLs because some of them are not working.
Author Response
Response: Thank you very much for the comments. We carefully revised the manuscript accordingly. The main revisions are marked in blue in the revised version.
- Check that the positions of the first and second cited articles in the citation list have not changed.
Response: Done. It has been corrected. Thank you for pointing out our mistake.
- The reference N 37 is an Inappropriate reference. This book makes no mention of the method you describe. See this articles: "Identification of nutrient deficiency in plants by artificial intelligence."
Response: Reference 37 has been revised to the one you suggested. The revised reference is reference 56. Thank you for your suggestion.
- Missing authors in reference N 46
Response: Reference 46 has been replaced with reference 66. Thank you for pointing out my error.
- Check the URLs because some of them are not working.
Response: Done.
Once again, thank you very much for your professional comments and suggestions. We earnestly appreciate your professional work and we hope that this revision will be deemed suitable for publication in Plants.
Reviewer 3 Report
The manuscript is highly recommended for bioinformatics study in plant breeding. The Authors try to remark on the analysis and application of bioinformatics in plant breeding.
The Authors should address some points:
1. Lines 51-53: Authors should add references in this sentence.
2. Lines 55-57: Authors should add references in this sentence.
3. Lines 58-89: Authors should add references in this sentence.
4. Lines 59-61: Authors should add references in this sentence.
5. Line 151: Authors should add references and discussion on the application of bioinformatics in chrysanthemum breeding.
6. Line 251: Authors should add references and discussion on the application of bioinformatics in research on chrysanthemum disease resistance.
Author Response
The manuscript is highly recommended for bioinformatics study in plant breeding. The Authors try to remark on the analysis and application of bioinformatics in plant breeding.
Response: Thank you very much for the comments. We carefully revised the manuscript accordingly. The main revisions are marked in blue in the revised version.
The Authors should address some points:
- Lines 51-53: Authors should add references in this sentence.
Response: Done. References have been added to this section; see lines 61-63 for details.
- Lines 55-57: Authors should add references in this sentence.
Response: References have been added to this section; see lines 66-68 for details.
- Lines 58-89: Authors should add references in this sentence.
Response: References have been added to this section; see lines 68-69 for details.
- Lines 59-61: Authors should add references in this sentence.
Response: References have been added to this section; see lines 69-72 for details.
- Line 151: Authors should add references and discussion on the application of bioinformatics in chrysanthemum breeding.
Response: Thanks for your suggestion, references and related discussions on the application of bioinformatics in chrysanthemum breeding have been added to this section. See lines 185-195 for details.
For example, in Chrysanthemum, GWASs has been used to explore genetic patterns and identify favorable alleles for several ornamental and resistance traits, including plant structural and inflorescence traits, waterlogging tolerance, aphid resistance, and drought tolerance [42]. Su et al. transferred a major SNP co-isolated with waterlogging tolerance in Chrysanthemum to a PCR-based derived cut amplified polymorphism sequence (dCAPS) marker with an accuracy of 78.9%, which was verified in 52 cultivars or progenitor [43]. Chong et al. developed two dCAPS markers associated with flowering stage and diameter of the head in Chrysanthemum. These dCAPS markers have potential applications in molecular breeding of Chrysanthemum [44]. These techniques will provide new powerful tools for future Chrysanthemum breeding.
- Line 251: Authors should add references and discussion on the application of bioinformatics in research on chrysanthemum disease resistance.
Response: Thank you for your suggestion. The discussion on the application of bioinformatics in chrysanthemum disease resistance research has been included in this section. Thanks again for your advice.
Chrysanthemum dwarf viroids (CSVd) can invade the leaf primordium and cells very close to the apical dome or even the outermost layer of the apical dome. The lack of CSVd in shoot tip seriously affected the asexual propagation of chrysanthemum [84]. Some researchers have developed a plant disease resistance protein predictor (RD-RFPDR), which showed to be sensitive and specific in identifying DR proteins after excluding data imbalance. This can provide a method for predicting plant disease resistance proteins [85].
Once again, thank you very much for your professional comments and suggestions. We earnestly appreciate your professional work and we hope that this revision will be deemed suitable for publication in Plants.
Round 2
Reviewer 1 Report
In general, the manuscript is improved compared to the previous version and more focused on the use of bioinformatics in the study of disease resistance in plants. However, some minor errors in editing and constructing the English language need to be corrected.
For example:
Lines 87-92, in which they explain the aims or focuses of the review.
If authors decide to list actions using the verb in the infinitive, they must keep this setting for all points:
In this review, we focus on the applications of bioinformatics in crop breeding and the study of resistance to various stress factors, to 1) list the applications of bioinformatics in plant research, 2) to clarify the application of bioinformatics in plant breeding, 3) to emphasize the advances made by bioinformatics in the study of plant tolerance and disease resistance 4) to predict the plant growth by bioinformatics; 5) to call for greater use of bioinformatic methods in plant research.
LL. 216-219. The sentence is imperative and is too colloquial. Please, modify. One example:
In the application it is possible to use the bioinformatics method to measure the stronger leaf angle of plant photosynthesis, from the perspective of bioinformatics analysis to create the optimal Angle of leaves, and to increase the accumulation of organic matter in plants.
LL. 267-268. The sentence is not correct, please correct:
………this new breeding method operation is simple, low cost, does not destroy the organism's own genes.
LL. 279-280. “In plant research, genotypic phenotypic prediction has traditionally used statistical methods.” Do you mean genotype-phenotype prediction, as in lll. 282-283?
In table 1, for the Meyers Lab database, use plant small RNA and not rna
Also, still pay attention to the list of references, because in some cases (eg. 87) thy are not complete yet.
After these revisions, I think the work may be accepted for publication.
Author Response
Comments: In general, the manuscript is improved compared to the previous version and more focused on the use of bioinformatics in the study of disease resistance in plants. However, some minor errors in editing and constructing the English language need to be corrected.
Response: Thank you very much again for these professional suggestions. These advices are constructive and helpful to greater improve our manuscript. We have carefully revised the manuscript accordingly. The main revisions are highlighted in red in the revised version (round 2). Detailed responses are listed below.
For example:
- Comments: Lines 87-92, in which they explain the aims or focuses of the review.
If authors decide to list actions using the verb in the infinitive, they must keep this setting for all points:
In this review, we focus on the applications of bioinformatics in crop breeding and the study of resistance to various stress factors, to 1) list the applications of bioinformatics in plant research, 2) to clarify the application of bioinformatics in plant breeding, 3) to emphasize the advances made by bioinformatics in the study of plant tolerance and disease resistance 4) to predict the plant growth by bioinformatics; 5) to call for greater use of bioinformatic methods in plant research.
Response: Done and thanks for these good advice. The details are revised in lines 87-92. In this review, we focus on the applications of bioinformatics in crop breeding and the study of resistance to various stress factors, to 1) list the applications of bioinformatics in plant research, 2) to clarify the application of bioinformatics in plant breeding, 3) to emphasize the advances made by bioinformatics in the study of plant tolerance and disease resistance 4) to predict the plant growth by bioinformatics; 5) to call for greater use of bioinformatic methods in plant research.
- Comments: LL. 216-219. The sentence is imperative and is too colloquial. Please, modify. One example:
In the application it is possible to use the bioinformatics method to measure the stronger leaf angle of plant photosynthesis, from the perspective of bioinformatics analysis to create the optimal Angle of leaves, and to increase the accumulation of organic matter in plants.
Response: Done. We have adopted your suggestion and revised the original text. For details, see lines 216-219. In the applications, it is possible to use the bioinformatics method to measure the stronger leaf angle of plant photosynthesis, from the perspective of bioinformatics analysis to create the optimal Angle of leaves, and to increase the accumulation of organic matter in plants.
- Comments: LL. 267-268. The sentence is not correct, please correct:
………this new breeding method operation is simple, low cost, does not destroy the organism's own genes.
Response: Thank you for the suggestions. We have corrected this sentence according to your advice shown in lines 268-269 for details “and this new breeding method operation is simple, low cost, does not destroy the organism's own genes.”.
- Comments: LL. 279-280. “In plant research, genotypic phenotypic prediction has traditionally used statistical methods.” Do you mean genotype-phenotype prediction, as in lll. 282-283?
Response: Yes, it's genotype-phenotype prediction. We have amended the detailed contents “In plant research, genotype-phenotype prediction has traditionally used statistical methods. For example, two statistical methods, autoregressive (AR) and Markov chain (MCMC), are used to predict the growth trends of plants by using normalized Vegetation Index (NDVI) [63].” shown in lines 280-283 for details.
- Comments: In table 1, for the Meyers Lab database, use plant small RNA and not rna
Response: Done and sorry for the mistake. Table 1 has been modified as “NGS database, including small RNAs and genome resources for plants” accordingly.
- Comments: Also, still pay attention to the list of references, because in some cases (eg. 87) thy are not complete yet.
Response: Thank you for your suggestion, Reference 87 has been revised and we have also checked all the reference lists.
- Comments: After these revisions, I think the work may be accepted for publication.
Once again, thank you very much for your professional comments and suggestions. We earnestly appreciate your professional work and we hope that this revision will be deemed suitable for publication in Plants.
Reviewer 3 Report
This manuscript was revised as the Reviewer suggested. It could be considered for publication in the MDPI journal.
Author Response
Response: Thank you very much again for your professional comments and suggestions. We earnestly appreciate your professional work and we hope that this revision will be deemed suitable for publication in Plants.